# Seed Nanopriming with ZnO and SiO_2_ Enhances Germination, Seedling Vigor, and Antioxidant Defense Under Drought Stress

**DOI:** 10.3390/plants14111726

**Published:** 2025-06-05

**Authors:** Erick H. Ochoa-Chaparro, Juan J. Patiño-Cruz, Julio C. Anchondo-Páez, Sandra Pérez-Álvarez, Celia Chávez-Mendoza, Luis U. Castruita-Esparza, Ezequiel Muñoz Márquez, Esteban Sánchez

**Affiliations:** 1Food and Development Research Center, A.C. Avenida Cuarta Sur No. 3820, Fraccionamiento Vencedores del Desierto, Delicias 33089, Chihuahua, Mexico; eriicktronik@hotmail.com (E.H.O.-C.); juanjose.industrias@gmail.com (J.J.P.-C.); anchondo_456@hotmail.com (J.C.A.-P.); celia.chavez@ciad.mx (C.C.-M.); emunoz@ciad.mx (E.M.M.); 2Faculty of Agricultural and Forestry Sciences, Universidad Autónoma de Chihuahua (UACH), Km. 2.5 Carretera a Rosales, Poniente, Delicias 33000, Chihuahua, Mexico; spalvarez@uach.mx (S.P.-Á.); lcastruita@uach.mx (L.U.C.-E.)

**Keywords:** drought stress, germination, jalapeño pepper, nanopriming, seedling, SiO_2_, ZnO

## Abstract

Drought stress is one of the main factors limiting seed germination and seedling establishment in field crops such as jalapeño peppers (*Capsicum annuum* L.). Nanopriming, a seed improvement technique using nanoparticle suspensions, has emerged as a sustainable approach to improving water use efficiency during the early stages of development. This study evaluated the effects of zinc oxide (ZnO, 100 mg·L^−1^), silicon dioxide (SiO_2_, 10 mg·L^−1^), and their combination (ZnO + SiO_2_), stabilized with chitosan, on the germination yield and drought tolerance of jalapeño seeds under mannitol-induced water stress (0%, 15%, and 30%). Compared to the hydroprimed control (T1), nanoparticle treatments consistently improved seed yield. Priming with ZnO (T2) increased the germination percentage by up to 25%, priming with SiO_2_ (T3) improved the germination rate by 34%, and the combined treatment (T4: ZnO + SiO_2_) improved the fresh weight of the seedlings by 40%. Proline accumulation increased 7.5 times, antioxidant capacity (DPPH) increased 6.5 times, and total phenol content increased 4.8 times in the combined treatment. Flavonoid levels also showed notable increases, suggesting enhanced antioxidant defense. These results clearly demonstrate the superior efficacy of nanoparticle pretreatment compared to conventional hydraulic pretreatment, especially under drought conditions. Multivariate analysis further highlighted the synergistic role of ZnO and SiO_2_ in improving osmolite accumulation, antioxidant activity, and water use efficiency. Nanopriming with ZnO and SiO_2_ offers a promising, economical, and scalable strategy to improve germination, early growth, and drought resistance in jalapeño pepper cultivation under semi-arid conditions.

## 1. Introduction

Drought is one of the most limiting abiotic factors affecting agricultural productivity worldwide, particularly in arid and semi-arid regions [1]. Water scarcity reduces seed germination rates, delays seedling establishment, and affects photosynthetic capacity while promoting the accumulation of reactive oxygen species (ROS), which cause oxidative stress, cellular damage, and membrane instability [2,3,4]. These physiological and biochemical alterations severely limit crop yields and threaten global food security, especially in the context of climate change [5,6]. Drought also disrupts root system architecture and water transport, exacerbating its impact on plant performance [7]. Given the increasing challenges posed by drought, it becomes essential to focus on crops of high nutritional and economic importance, particularly those cultivated in water-limited environments.

One such crop is jalapeño pepper (*Capsicum annuum* L.), which holds significant nutritional and economic value, especially in Mexico, where it is both a staple food and an important export commodity [8,9]. Jalapeño fruits are rich in bioactive compounds such as capsaicinoids, flavonoids, and vitamins A, C, and E, known for their antioxidant properties and health benefits [10,11]. Mexico ranks as the second largest global producer of jalapeño, with over 50% of its production concentrated in semi-arid northern regions where water scarcity is a major constraint [12]. This combination of nutritional importance and vulnerability to drought highlights the need for innovative approaches to improve the resilience of jalapeño under water-limited conditions [13,14].

Among the strategies developed to improve crop resilience, seed nanopriming has emerged as a promising technique. This approach involves treating seeds with nanoparticle suspensions, allowing for the uptake of active compounds that can stimulate physiological and biochemical responses to stress [15,16]. Zinc oxide (ZnO) and silicon dioxide (SiO_2_) nanoparticles have demonstrated potential to enhance osmotic regulation, antioxidant defense, and seedling vigor under drought conditions in various crops, including wheat, maize, and tomato [17,18,19,20]. However, the potential of combining ZnO and SiO_2_ nanopriming in jalapeño pepper under simulated drought stress has not been thoroughly investigated, leaving a gap in our understanding of its effects on key physiological and biochemical parameters.

To evaluate the effectiveness of such treatments, it is essential to consider key response variables in plants subjected to drought stress. These include morphological traits such as germination percentage, seedling length, and biomass, as well as biochemical markers like proline, phenolics, and flavonoids [4,21,22]. These metabolites are crucial for osmotic adjustment, ROS detoxification, and membrane stabilization, playing a central role in the plant’s defense mechanisms under water deficit conditions [2,5,16]. By exploring how nanopriming modulates these variables in jalapeño pepper, it is possible to develop targeted strategies that enhance drought resilience in this important crop.

Therefore, the objective of this study was to evaluate the effectiveness of seed nanopriming with ZnO, SiO_2_, and their combination in improving the germination, seedling growth, and antioxidant responses of jalapeño pepper under mannitol-induced drought stress. This study aims to provide practical, scalable solutions to enhance drought tolerance in jalapeño cultivation, promoting sustainable agricultural practices in semi-arid and water-limited environments.

## 2. Results

### 2.1. Germination Parameters

Seed germination was significantly influenced by nanopriming treatments, drought stress levels, and jalapeño pepper varieties (*p* ≤ 0.05), as shown in Table 1. Treatments were coded using a combination of the variety abbreviation, treatment number, and stress level (see Table 2). For example, MT215 corresponds to the Mixteco variety treated with ZnO (Treatment 2) under 15% drought stress.

Under optimal conditions (0%), treatments such as IMT10, FT10, IMT20, FT20, IMT30, and IMT40 achieved 100% germination, while IT10 (Ideal + Hydropriming) showed the lowest germination (75%). Nanopriming with SiO_2_ (IT30) resulted in the shortest average germination time (2.33 ± 0.08 days) and the highest germination rate (0.429 ± 0.015 day^−1^), outperforming FT30 (4.13 ± 0.14 days and 0.242 ± 0.009 day^−1^, respectively).

Under severe drought conditions (30%), FT230 and MT330 achieved complete germination, while IMT430 and FT430 showed reduced performance (79.16%). Notably, FT230 also had the lowest uncertainty index (0.86 ± 0.08), indicating greater synchronization in germination events.

### 2.2. Morphological Parameters

Nanopriming significantly improved seedling morphology, especially under water stress (Figure 1 and Figure 2). With 0% drought, FT10 recorded the longest shoot length (27 ± 0.91 mm) and the highest fresh weight (68.67 ± 4.16 mg), while FT20 and MT30 showed the longest roots. MT30 also had the largest stem diameter (0.3 ± 0.1 mm), three times larger than that of IMT30.

With 15% stress, FT415 (Forajido + ZnO + SiO_2_) produced the highest fresh weight of the seedlings, while IT415 had the longest shoots. FT315 showed the most extensive root growth (87.5 ± 2.88 mm). The highest shoot-to-root ratio was observed in IMT215 (0.54), while FT115 had the lowest (0.13), indicating a variation in resource allocation among the treatments.

Under 30% water stress, IT130 had the longest shoots (16.3 ± 3.1 mm), and MT330 and MT130 had the longest roots. IMT330 and MT130 also had the highest fresh weights (>34 mg). IT230 showed the highest shoot-to-root ratio (1.56), suggesting preferential shoot development under extreme drought conditions.

### 2.3. Biochemical Parameters

#### 2.3.1. Photosynthetic Pigments

Pigment levels (chlorophyll a, chlorophyll b, total chlorophyll, and carotenoids) were analyzed using two-way ANOVA (water stress level x priming treatment) performed separately for each jalapeño variety, followed by Tukey’s HSD test (*p* ≤ 0.05).

Chlorophyll and carotenoid content varied significantly between treatments and stress levels (Figure 3A–D). In the absence of water stress (0%), the Mixteco variety with the ZnO + SiO_2_ treatment (MT40) had the highest values of total chlorophyll (787.26 ± 31.79 µg·g^−1^ PF) and carotenoids (103.49 ± 8.25 µg·g^−1^ PF). Under moderate stress (15%), IT415 (Ideal + ZnO + SiO_2_) achieved the highest total chlorophyll content (831.64 ± 173.63 µg·g^−1^ PF). Under severe stress conditions (30%), IT130 (Ideal + Hydropriming) showed the highest concentrations in all pigment variables, indicating remarkable stress tolerance under that specific treatment.

#### 2.3.2. Soluble Sugars

Glucose, fructose, and sucrose levels were analyzed using two-way ANOVA (drought stress level x priming treatment) performed separately for each jalapeño variety, followed by Tukey’s HSD test (*p* ≤ 0.05). Sugar content significantly increased in response to both drought stress and nanopriming treatments (Figure 3E–G).

Under non-stress conditions (0% drought), the Forajido variety treated with ZnO + SiO_2_ (FT40) accumulated the highest total sugar content (24.39 mg·g^−1^ FW). Under 15% drought stress, IMT415 (Imperial + ZnO + SiO_2_) exhibited significantly elevated levels of glucose, fructose, and sucrose. Under 30% drought stress, IMT230 (Imperial + ZnO + Q) recorded the highest sugar accumulation.

#### 2.3.3. Free Proline

Proline content showed significant variation between treatments and drought levels (*p* ≤ 0.05), as shown in Figure 4A. Nanopriming with ZnO, especially when combined with SiO_2_, resulted in substantial proline accumulation.

Under stress-free conditions (0%), FT40 (Forajido + ZnO + SiO_2_) recorded the highest proline level (790.29 ± 254.64 µg·g^−1^ FW), more than 50 times higher than that of IT10 (Ideal + hydropriming), which had the lowest value. With 15% drought, FT415 showed the highest accumulation (410.36 ± 104.89 µg·g^−1^ FW), while IT315 had the lowest (22.64 ± 18.88 µg·g^−1^ FW). Under severe stress (30%), IT430 surpassed all treatments with 448.50 ± 189.00 µg·g^−1^ FW, compared to only 53.08 ± 8.61 µg·g^−1^ FW in IT130. These findings confirm the role of proline in osmotic adjustment and indicate that ZnO and ZnO + SiO_2_ priming effectively promote stress tolerance through osmoprotective mechanisms.

#### 2.3.4. Total Phenolic Content (TPC)

Significant differences in TPC were detected between treatments and stress levels (Figure 4B). With 0% stress, FT40 (Forajido + ZnO + SiO_2_) had the highest TPC (55.28 ± 5.89 mg GAE·g^−1^ FW), almost 4.8 times higher than that of MT10 (Mixteco + Hydropriming), which had the lowest content (11.50 ± 0.41 mg GAE·g^−1^ FW).

With 15% stress, IT115 (Ideal + Hydropriming) showed the highest TPC (26.41 ± 3.68 mg GAE·g^−1^ FW), while FT215 (Forajido + ZnO) had the lowest (10.81 ± 0.89 mg GAE·g^−1^ FW). With 30% drought, IT430 maintained the highest phenolic content (51.40 ± 6.35 mg GAE·g^−1^ FW), representing a 3.2-fold increase compared to IT130 (16.08 ± 10.45 mg GAE·g^−1^ FW). These data suggest that the combination of ZnO and SiO_2_ increases phenol accumulation, especially under severe drought conditions.

#### 2.3.5. Total Flavonoid Content (TFC)

TFC also showed differences depending on treatment and stress (Figure 4C). Under optimal conditions, FT40 recorded the highest TFC (21.97 ± 8.40 mg QE·g^−1^ FW), almost 100 times higher than that of IT40 (0.22 ± 0.13 mg QE·g^−1^ FW). Under 15% stress, FT115 had the highest TFC (9.00 ± 1.90 mg QE·g^−1^ FW), while IT315 had the lowest (0.14 ± 0.67 mg QE·g^−1^ FW). With 30% drought, FT430 retained the highest flavonoid levels (16.20 ± 5.03 mg QE·g^−1^ FW), 70 times higher than those of FT230. These results highlight the improvement in flavonoid biosynthesis caused by ZnO and SiO_2_, especially in combined treatments.

#### 2.3.6. DPPH Antioxidant Activity

Antioxidant activity, evaluated by DPPH (2,2-difenil-1-picrilhidrazil) radical scavenging, varied significantly among treatments and stress levels (*p* ≤ 0.05) (Figure 4D). With 0% stress, FT40 showed the highest activity (15.72 ± 1.01 mg TE·g^−1^ FW), indicating a strong baseline antioxidant potential. Under 15% stress, IT415 showed the highest DPPH activity (5.61 ± 0.01 mg TE·g^−1^ FW), while under 30% drought, IT430 again led the way (11.53 ± 0.63 mg TE·g^−1^ FW). These results suggest that nanopriming with ZnO + SiO_2_ effectively stimulates antioxidant defense systems over a wide range of drought intensities.

### 2.4. PCA and Heatmap Analyses

#### 2.4.1. PCA and Heatmap Under 0% Stress

Under stress-free conditions, principal component analysis (PCA) explained 55.23% of the total variance between treatments. PC1 accounted for 33.56% and was mainly associated with morphological traits such as shoot and root length, fresh weight, and chlorophyll content. PC2 explained 21.67% of the variance and was related to biochemical parameters such as proline, total flavonoid content (TFC), and DPPH activity (Figure 5A). This separation indicates that, even in the absence of drought, nanopriming influences growth and antioxidant responses differently.

The heatmap (Figure 5B) reveals strong positive correlations between total chlorophyll content and phenolic content (TPC), as well as DPPH activity (r > 0.87), indicating a functional relationship between photosynthetic pigments and antioxidant capacity under stress-free conditions.

#### 2.4.2. PCA and Heatmap Under 15% Stress

With a water stress of 15%, PCA explained 36.61% of the total variation. PC1 grouped morphological traits such as germination speed, shoot length, fresh weight, and vigor indices. PC2 was mainly driven by variables related to antioxidants, such as proline, TFC, and DPPH (Figure 5C). This suggests that nanopriming under moderate drought conditions triggers an integrated physiological and biochemical response.

The corresponding heatmap (Figure 5D) highlights a strong clustering of proline, DPPH, and TFC, with correlation coefficients above 0.85. This indicates a coordinated response of osmoprotective and antioxidant pathways under moderate stress.

#### 2.4.3. PCA and Heatmap Under 30% Stress

Under severe water stress conditions (30% mannitol), PCA accounted for 40.74% of the total variance. PC1 captured most of the variation and was dominated by osmoprotective and antioxidant variables—proline, soluble sugars, TPC, and DPPH activity—reflecting a metabolic shift toward biochemical protection. PC2 was largely associated with photosynthetic pigments (chlorophyll a, b, and carotenoids) (Figure 5E).

The heatmap (Figure 5F) shows very strong correlations between proline, DPPH, and TPC (r > 0.90), supporting the hypothesis that responses to oxidative and osmotic stress are closely coordinated under high water stress. In addition, chlorophylls and carotenoids showed a high mutual correlation (r = 0.89), indicating synchronized photoprotective mechanisms.

These multivariate patterns demonstrate that the response to drought stress varies depending on the treatment and severity level, and that priming with ZnO and SiO_2_, especially in combination, is consistently associated with stress resistance traits.

### 2.5. Radar Chart: Multivariate Comparison by Drought Level

A radial graph (Figure 6) was used to visualize the comparative performance of nanopriming treatments on 15 key variables and three stress levels (0%, 15%, and 30%). Each axis represents a trait, and each line corresponds to a treatment: ZnO (green), SiO_2_ (red), and ZnO + SiO_2_ (blue).

Under optimal conditions (0%), ZnO led in carotenoids, shoot length, and total chlorophyll, while SiO_2_ + ZnO outperformed the others in phenols and DPPH activity. SiO_2_ alone improved fresh weight and sugar content, indicating osmotic benefits even in stress-free environments.

With 15% stress, SiO_2_ treatment consistently outperformed the others in germination speed, DPPH activity, phenol accumulation, and fresh biomass. ZnO was favorable for vigor and shoot–root balance, while the combined treatment showed a balanced profile in all biochemical and morphological parameters.

With 30% stress, the combination of ZnO + SiO_2_ showed superior performance in total chlorophyll, DPPH activity, and TPC, indicating a strong photoprotective and antioxidant response. ZnO improved root development and germination, while SiO_2_ stood out in proline accumulation and TPC, reinforcing its role in osmotic adjustment and biochemical defense.

Overall, the radar chart illustrates that nanopriming combined with ZnO and SiO_2_ consistently elicits synergistic responses under drought conditions, offering a robust strategy for improving drought resistance in jalapeño pepper seedlings.

## 3. Discussion

This study demonstrated that nanopriming with ZnO, SiO_2_, and their combination effectively improved germination dynamics, early growth, and biochemical responses in jalapeño peppers (*Capsicum annuum* L.) under different levels of drought stress. The responses observed depended on the type of nanoparticle, the severity of water stress, and the plant variety, highlighting the relevance of interactions between variety, environment, and nanomaterials.

These results support previous findings indicating that nanopriming can improve crop performance under abiotic stress by triggering physiological and biochemical responses that enhance water use efficiency, osmotic regulation, and oxidative stress management [16,21,22].

Under optimal conditions (0% water stress), priming with ZnO and SiO_2_ significantly improved germination parameters. Treatments such as IMT20, FT20, and IMT40 promoted faster and more uniform germination compared to the hydroprimed control. This improved performance can be attributed to increased α-amylase activity and gibberellin signaling, which accelerate starch degradation and reserve mobilization, favoring early seedling development [22]. The significant reduction in average germination time and uncertainty in IT30, a SiO_2_ priming treatment, suggests early metabolic activation potentially related to membrane stabilization and increased aquaporin activity [23,24]. Even without water stress, the activation of these physiological mechanisms through nanopriming likely creates a “priming state”, which may confer long-term resilience.

Under moderate water stress (15% mannitol), nanoprimed seeds in treatments such as FT415, IT315, and MT315 maintained high germination rates—up to 100% in FT415 and IT315—and showed vigorous root elongation, shoot development, and biomass accumulation. These responses suggest that nanopriming facilitates the maintenance of cell turgidity, osmotic balance, and early stress signal transduction. This is consistent with previous findings in chickpeas, tomatoes, and okra, where ZnO and SiO_2_ NPs improved drought tolerance by maintaining root growth and physiological function [25,26,27,28]. Specifically, ZnO-treated seedlings showed improved root architecture, likely due to increased mitotic activity and root system plasticity, which allowed for greater water uptake under conditions of limited availability [29]. In addition, ZnO contributed to the balance between shoots and roots, while SiO_2_ improved osmotic regulation and preserved tissue integrity by reinforcing cell wall stability and hydration dynamics [30].

Under severe water stress (30% mannitol), although biomass accumulation decreased, treatments such as FT430, IT430, and MT230 maintained high germination rates and showed strong accumulation of osmoprotectants and antioxidant compounds. This shift from growth promotion to biochemical defense enhancement indicates a physiological balance that prioritizes survival over expansion, a well-established adaptive response under abiotic stress [31]. It is noteworthy that proline levels increased dramatically, especially in ZnO+SiO_2_ treatments, confirming its role as an osmoprotectant and ROS (reactive oxygen species) scavenger [20].

Proline also showed strong correlations with phenols and DPPH activity, consistent with stress responses in *Mentha pulegium* and corn [32,33]. Similar synergistic effects of silicon and zinc NPs in improving osmoprotectant accumulation and stress mitigation in wheat under salinity have been reported, where the joint application of proline with Zn and Si NPs significantly improved proline content, antioxidant activity, and plant growth [34]. These results reinforce the concept that, under conditions of intense drought, nanopriming modulates seedling metabolism towards greater biochemical resilience rather than growth, particularly through the synergistic effects of ZnO and SiO_2_.

Similarly, total phenolic content (TPC) and total flavonoid content (TFC) increased significantly in nanoparticle treatments, particularly FT40 and IT430. These secondary metabolites play an essential role in non-enzymatic antioxidant defense, redox regulation, and cell protection under stress. Their accumulation under optimal and drought conditions suggests that nanopriming may induce a mild oxidative signal that pre-activates antioxidant pathways, improving the plant’s preparedness for stress [21]. This state of preparedness was supported by the high DPPH radical scavenging activity in the same treatments, indicating a robust antioxidant response. These findings are consistent with previous studies showing that ZnO and SiO_2_ NPs improve phenolic biosynthesis and antioxidant potential in crops such as potato and wheat under water-limited conditions [34,35,36]. Furthermore, evidence obtained with NPs based on Mimusops elengi extracts also supports their role in stimulating phenol production and ROS detoxification [37]. In line with this, the antioxidant effects observed here are consistent with reports on plant systems treated with pink pepper and citrus extracts under oxidative conditions [38].

Multivariate analyses further confirmed the integrative role of nanopriming in modulating growth and defense responses under water stress. Principal component analysis (PCA) revealed that, under 0% stress, germination and morphological traits—such as root length, shoot length, fresh weight, and chlorophyll content—had a large weight in PC1, while antioxidant markers such as proline, total flavonoid content (TFC), and DPPH activity contributed predominantly to PC2. This clear separation reflects the dual action of nanopriming in promoting early vigor and activating defense pathways. As drought intensified to 15% and 30% mannitol, the PCA structure changed markedly: osmoprotectants and antioxidants explained most of the variance, indicating a physiological transition toward biochemical prioritization. Similar patterns have been described in chickpea, where PCA-based multivariate selection was driven by biochemical markers under stress conditions [39]. Under 30% stress, variables such as proline and glucose became the main contributors to PC1, highlighting their central role in osmotic adjustment and metabolic reprogramming.

Heatmap correlation analysis complemented these findings, revealing strong associations (r > 0.90) between proline, DPPH, phenols, and TFC, especially under moderate and severe stress, indicating a closely coordinated antioxidant and osmotic defense network. These trends mirror observations made in Cicer arietinum and maize, where ROS scavenging, osmotic balance, and root system architecture (RSA) were crucial for drought adaptation [26,40]. Under low stress, high correlations between chlorophylls, phenols, and DPPH reflected physiological homeostasis. Under 15% stress, the synergy between TFC and proline with fresh weight suggested metabolic flexibility and maintenance of turgidity. At 30%, increased correlations between sugars and antioxidant compounds revealed a trade-off favoring stress survival over growth, a phenomenon described as an adaptation strategy in plants under drought stress [31].

To better visualize the multidimensional performance of each treatment, a radial graph (Figure 6) was constructed that summarizes 15 variables at all stress levels. This graph provided an intuitive and comprehensive view of the relative strengths of each nanopriming treatment. Under 0% stress, ZnO treatments stood out in photosynthetic pigment content and shoot development, while SiO_2_ + ZnO showed superior antioxidant capacity. SiO_2_ alone improved fresh weight and sugar content, highlighting its role in osmotic stabilization even in unstressed seedlings. Under 15% stress, SiO_2_ consistently obtained the best results in germination speed, antioxidant activity, phenol accumulation, and biomass, suggesting strong ROS detoxification and greater water use efficiency. ZnO was most effective in maintaining the balance between shoots and roots, and the combined ZnO + SiO_2_ treatment showed a balanced profile in morphological and biochemical dimensions.

Under severe drought conditions (30%), the superiority of the ZnO+SiO_2_ treatment became even more evident. This combined treatment outperformed all others in total chlorophyll, DPPH activity, and phenols, reflecting integrated photoprotective and antioxidant defense. ZnO favored root development and germination under stress, while SiO_2_ treatments achieved the highest levels of proline, further reinforcing its osmoprotective role. Thus, the radial graph confirmed the complementarity and potential synergism between these NPs, especially under difficult environmental conditions. It also illustrated the ability of nanopriming to activate different but interconnected response mechanisms depending on the level of drought.

Together, these results underscore the value of nanopriming as a specific, economical, and scalable technology for improving crop performance under drought conditions. In regions such as northern Mexico, where more than 50% of the national jalapeño production occurs and water scarcity is a major constraint [13,41], the implementation of nanopriming protocols could significantly improve seedling establishment and productivity. Furthermore, the consistency of responses across varieties and stress levels suggests that these technologies could be extended to other crops and production systems.

Together, the evidence indicates that nanopriming with ZnO, SiO_2_, and their combination plays a crucial role in improving the physiological and biochemical performance of jalapeño pepper seedlings under drought conditions. These treatments promoted more efficient germination, improved early structural development, and activated antioxidant defense pathways, especially under moderate and severe water deficit conditions. Integrative analysis using a radar chart highlighted the multidimensional advantages of the combined ZnO+SiO_2_ treatment, reinforcing its value as a synergistic and promising approach for improving drought resilience during the most vulnerable stages of plant establishment.

When evaluating the overall efficacy of the treatments, a clear trend in the order of efficacy under drought stress conditions was observed: Hydropriming ≤ SiO_2_ ≤ ZnO ≤ ZnO + SiO_2_. Hydropriming treatment showed the least improvement in most variables, indicating a limited ability to improve drought resistance. SiO_2_ priming improved certain parameters, such as antioxidant activity, but its effects were less consistent compared to ZnO. ZnO treatment demonstrated superior performance in promoting osmotic adjustment, antioxidant responses, and early growth. In particular, the combined ZnO + SiO_2_ treatment consistently outperformed the individual treatments, showing a synergistic effect that maximized physiological and biochemical improvements in seedling germination and development under water deficit conditions. These findings highlight the potential of ZnO + SiO_2_ nanopriming as the most effective strategy for improving drought tolerance in jalapeño peppers.

## 4. Materials and Methods

### 4.1. Experimental Conditions and Plant Material

The experiment was conducted from 24 October to 17 December 2024 at the Food and Development Research Center (CIAD), Delicias Unit, Chihuahua, Mexico. Four commercial varieties of jalapeño pepper F1 (*Capsicum annuum* L.) were used: Mixteco, Ideal, Imperial, and Forajido. The seeds were obtained from Invernaderos Ramos, located in Lomas del Consuelo, Meoqui, Chihuahua, Mexico. The seeds were visually inspected to ensure uniformity and then disinfected with 4% sodium hypochlorite for 2 min, followed by three rinses with sterile distilled water, as described by Pandya et al. [42].

### 4.2. Nanoparticle Preparation and Priming Treatments

The concentrations of ZnO (100 mg L^−1^) and SiO_2_ (10 mg L^−1^) used in this study were selected based on previous studies that demonstrated effective physiological responses in crops under drought stress conditions [17,18]. These doses have been reported to improve osmotic adjustment, antioxidant defense, and seedling growth without causing toxicity to plants.

The zinc oxide (ZnO) and silicon dioxide (SiO_2_) nanoparticles (NPs) used in this study were synthesized by using the wet chemical method. For ZnO, a Wurtzite-type crystalline structure was obtained with a purity of 99.7% and an average particle size of 50 nm, free of contaminants. For SiO_2_, a purity of 99.9% was obtained, with a physical appearance of a fine, light-colored powder and an average particle size of 80 nm. Both NPs were structurally and morphologically characterized by transmission electron microscopy (TEM) (Figure 7), confirming their homogeneity and stability. The nanoparticles were provided by the company Investigación y Desarrollo de Nanomateriales S.A. de C.V., located in San Luis Potosí, Mexico.

### 4.3. Characterization of Zinc Oxide and Silicon Dioxide Nanoparticles

Zinc oxide (ZnO) and silicon dioxide (SiO_2_) NPs were synthesized using the coprecipitation method and characterized by transmission electron microscopy (TEM), which confirmed an average size of 50–80 nm (Figure 7). A solution of 100 mg L^−1^ of each nanoparticle was prepared in triple-distilled water with 0.1% chitosan as a dispersant. The solutions were homogenized by magnetic stirring (60 min) and sonication (15 min), as described by Waqas et al. [22]. The priming treatments consisted of soaking the seeds in 30 mL of each solution for 12 h at 25 °C in the dark, followed by drying at room temperature for 24 h [22]. The treatments were coded according to variety (M: Mixteco, I: Ideal, IM: Imperial, F: Forajido), nanoparticle type (T1: Hydropriming, T2: ZnO, T3: SiO_2_, T4: ZnO + SiO_2_), and stress level (10 = 0%, 115 = 15%, 130 = 30%) (see Table 1). These codes were used consistently in data collection, figures, and analysis (see Figure 1, Figure 2, Figure 3, Figure 4, Figure 5 and Figure 6).

### 4.4. Experimental Design

A completely randomized design (CRD) was used, following a 4 × 4 × 3 factorial arrangement, where factor A corresponded to four commercial varieties of jalapeño pepper F1 (*Capsicum annuum* L.): Mixteco, Ideal, Imperial, and Forajido; factor B consisted of four priming treatments (Hydropriming, ZnO, SiO_2_, and ZnO + SiO_2_); and factor C consisted of three levels of induced water stress (0%, 15%, and 30%) (Figure 8). Each experimental unit consisted of a 9 cm diameter Petri dish lined with Whatman No. 4 filter paper, moistened with 5 mL of the corresponding solution. For the control treatment, triple-distilled water was used to ensure no osmotic stress was applied. For the stress treatments, mannitol solutions at 15% (−0.24 MPa) and 30% (−0.48 MPa) were used, following the method described by Ertuş and Yazıcılar [43], as these levels simulate mild to moderate drought conditions commonly used in germination studies.

Eight seeds were placed in each Petri dish, and each seed was considered a replicate within its treatment. During the 8-day germination period, the Petri dishes were maintained in a controlled growth chamber under constant conditions of 28 ± 2 °C, 60% relative humidity, and a 16/8 h light/dark photoperiod. The filter papers were rehydrated with the corresponding solutions (distilled water or mannitol) as needed to maintain moisture levels throughout the experimental period.

The effects of nanopriming and water stress levels on germination, initial growth, chlorophyll content, total soluble carbohydrates, free proline, total phenols, total flavonoids, and antioxidant capacity evaluated by DPPH radical scavenging were assessed. Each analysis was performed in triplicate.

### 4.5. Germination Assay

Germination was monitored daily for 8 days. Seeds were considered germinated when the radicle exceeded 1 mm (Figure 9). Germination indices (percentage, mean germination time, mean germination rate, speed, and uncertainty) were calculated using the GerminaR package [44]. Germination evaluation followed the ISTA [45] guidelines. The results are shown in Table 1.

### 4.6. Morphological Measurements

On day 8, seedlings were evaluated for shoot length, root length, fresh weight, and stem diameter using a digital caliper. Radicle length was measured from the base of the hypocotyl to the apex of the radicle, while plumule (shoot) length was measured from the radicle–hypocotyl junction to the base of the cotyledons [41]. Stem diameter was measured at approximately 2 mm above the root–hypocotyl transition zone. Vigor indices I and II were calculated according to Abdul-Baki and Anderson [46].

Morphological variability between varieties was interpreted using the descriptors proposed by Elizondo-Cabalceta and Monge-Pérez [47]. Data were collected by variety and treatment to evaluate growth performance under different drought stress levels (Figure 2 and Figure 3).

### 4.7. Biochemical Analyses

Photosynthetic pigments (chlorophyll a, b, total chlorophyll, and carotenoids) were extracted with 99% methanol and quantified spectrophotometrically at 665, 652, and 470 nm, using the equations of Wellburn [48]. The proline content was quantified using the ninhydrin method, using benzene extraction and absorbance at 520 nm. Soluble sugars (glucose, fructose, sucrose) were extracted with 80% ethanol and analyzed using the anthrone method at 620 nm [49].

The total phenolic content (TPC) was determined using the Folin–Ciocalteu method with gallic acid as a standard [50], and the total flavonoid content (TFC) was measured using the AlCl₃ colorimetric method with quercetin as a standard [38]. Antioxidant activity was determined by DPPH radical scavenging [35]. The absorbance for TPC, TFC, and DPPH was measured at 765, 510, and 517 nm, respectively. The data for these biochemical traits are shown in Figure 3 and Figure 4.

### 4.8. Multivariate and Correlation Analysis

Principal component analysis (PCA) and Pearson correlation matrices were performed using OriginPro 2025 to evaluate relationships between morphological and biochemical traits. PCA biplots were generated separately for each drought level (0%, 15%, 30%) (Figure 5A,C,E), and heatmaps were constructed to visualize correlations between traits and treatments (Figure 5B,D,F) [39]. A radar chart (Figure 6) was created to integrate the germination, growth, and biochemical performance of ZnO, SiO_2_, and ZnO+SiO_2_ treatments across the three stress levels, normalizing 15 key variables.

### 4.9. Statistical Analysis

Each variable was represented by three replicates. Germination parameters were analyzed using the nonparametric Kruskal–Wallis test followed by Dunn’s post hoc test (*p* ≤ 0.05), due to the non-normal distribution of the data.

For early growth variables and biochemical parameters such as photosynthetic pigments and soluble sugars, the assumptions of normality, homogeneity of variances, and independence were tested. When these assumptions were met, two-way ANOVA (drought stress level × priming treatment) was performed separately by variety. Tukey’s HSD test (*p* ≤ 0.05) was used to determine significant differences between means.

For antioxidant-related variables—total phenols, flavonoids, proline content, and DPPH radical scavenging activity—three-way ANOVA was conducted to evaluate the effects of priming treatment, drought stress level, and variety, including their interactions. Post hoc comparisons were performed using Tukey’s HSD test (*p* ≤ 0.05).

All statistical analyses were carried out using SAS^®^ software version 9.0 (SAS Institute Inc., Cary, NC, USA [51]). Figures, bar charts, radar charts, Pearson correlation matrices, and principal component analysis (PCA) plots were generated using OriginPro 2025 (v10.2.0.196).

## 5. Conclusions

Nanopriming with ZnO, SiO_2_, and their combination significantly improved drought tolerance in jalapeño pepper seedlings by enhancing germination rates, seedling vigor, and antioxidant capacity under moderate and severe water stress conditions. The combined treatment with ZnO + SiO_2_ consistently produced superior results across all morphological and biochemical parameters, suggesting a synergistic effect in promoting stress resistance during early plant development. These findings demonstrate that seed nanopriming is an effective and scalable pre-planting strategy to optimize seed performance, improve osmotic regulation and antioxidant defenses, and mitigate the detrimental effects of water scarcity on crop establishment. Ultimately, this approach contributes to improving water use efficiency and supports the development of more resilient and sustainable agricultural systems in arid and semi-arid regions. 

## Figures and Tables

**Figure 1 plants-14-01726-f001:**
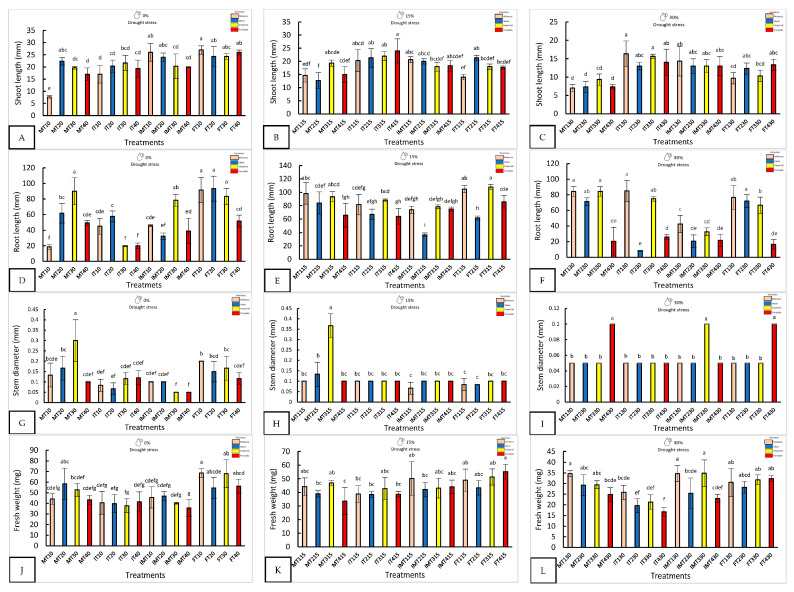
The morphological response of four varieties under nanopriming and drought stress. Panels (**A**–**L**) show shoot length, root length, stem diameter, and fresh weight in four jalapeño varieties (Mixteco, Ideal, Imperial and Forajido) under three drought levels (0%, 15% and 30%) and four treatments: T1 (Hydropriming), T2 (ZnO + Q), T3 (SiO_2_ + Q), and T4 (ZnO + SiO_2_ + Q). Bars represent the mean ± SD. Different letters indicate significant differences (Tukey HSD, *p* ≤ 0.05).

**Figure 2 plants-14-01726-f002:**
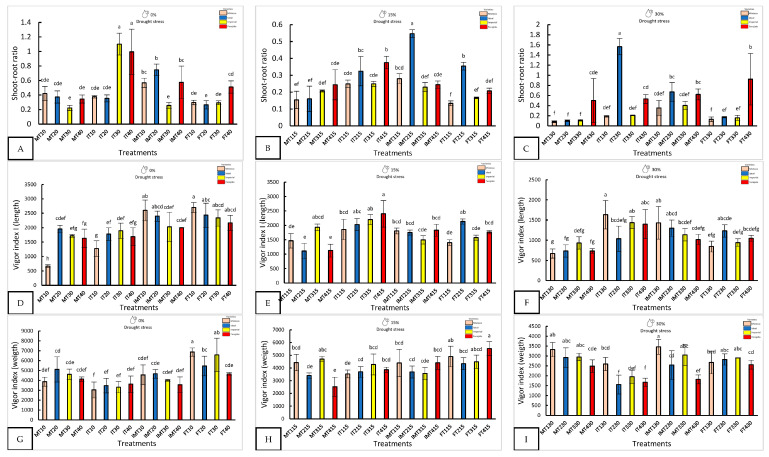
The morphological response of four varieties under nanopriming and drought stress. Panels (**A**–**I**) show the shoot–root ratio, Vigor Index I, and Vigor Index II in four jalapeño varieties (Mixteco, Ideal, Imperial and Forajido) under three drought levels (0%, 15%, and 30%) and four treatments: T1 (Hydropriming), T2 (ZnO + Q), T3 (SiO_2 +_ Q), and T4 (ZnO + SiO_2_ + Q). Bars represent the mean ± SD. Different letters indicate significant differences (Tukey HSD, *p* ≤ 0.05).

**Figure 3 plants-14-01726-f003:**
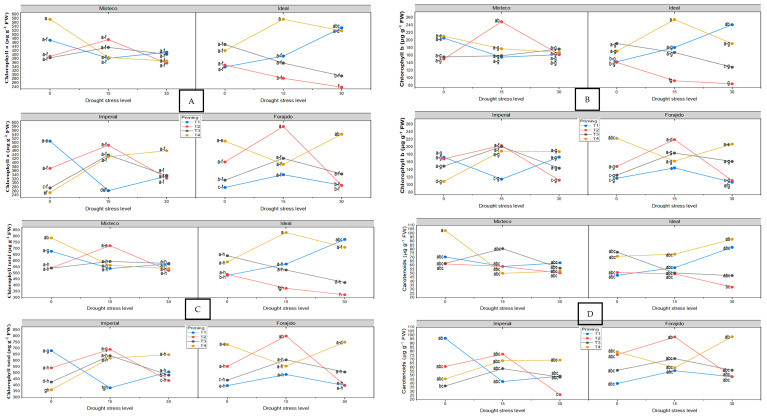
The effect of nanopriming treatments and drought stress levels on the photosynthetic pigments and soluble sugar composition of four jalapeño pepper varieties. Panels (**A**–**G**) display the response in the varieties Mixteco, Ideal, Imperial, and Forajido under three drought stress levels (0%, 15%, and 30%) and four nanopriming treatments: T1 (Hydropriming), T2 (ZnO + Q), T3 (SiO_2_ + Q), and T4 (ZnO + SiO_2_ + Q). The variables assessed were (**A**) chlorophyll (a); (**B**) chlorophyll (b); (**C**) total chlorophyll; (**D**) glucose; (**E**) fructose; (**F**) sucrose. Each data point represents the mean ± standard deviation. Different letters indicate statistically significant differences (Tukey HSD, *p* ≤ 0.05). Under 15% water stress, MT215 (Mixteco + ZnO) improved germination dynamics by reducing the average germination time by 26% and increasing the germination rate by 34% compared to IMT115. Several treatments, such as MT115, IT315, and IT415, maintained 100% germination even under water stress.

**Figure 4 plants-14-01726-f004:**
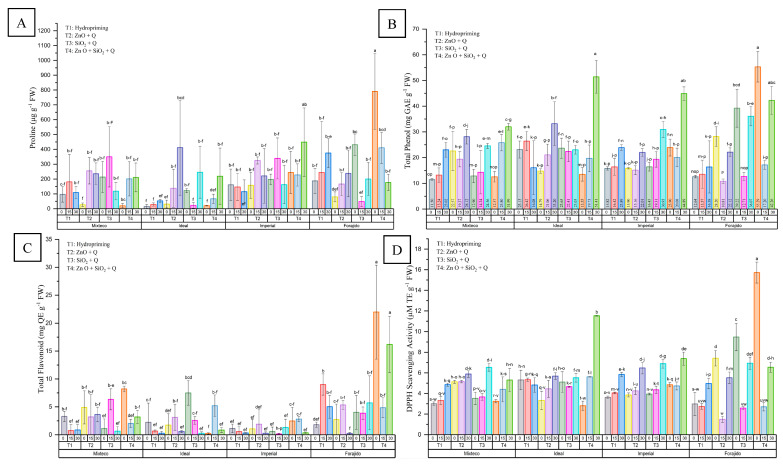
Biochemical responses in four jalapeño pepper varieties under different nanopriming treatments and drought stress levels. The panels show (**A**) free proline content (μg g^−1^ FW), (**B**) total phenolic content (mg GAE g^−1^ FW), (**C**) total flavonoid content (mg QE g^−1^ FW), and (**D**) DPPH radical scavenging activity (mg TE g^−1^ FW). Bars represent the mean ± standard deviation for each variable. The treatments applied were T1 (Hydropriming), T2 (ZnO + Q), T3 (SiO_2_ + Q), and T4 (ZnO + SiO_2_ + Q) across three drought stress levels (0%, 15%, and 30%) in the varieties Mixteco, Ideal, Imperial, and Forajido. Letters above the bars indicate significant differences between treatments according to a Tukey HSD test (*p* ≤ 0.05).

**Figure 5 plants-14-01726-f005:**
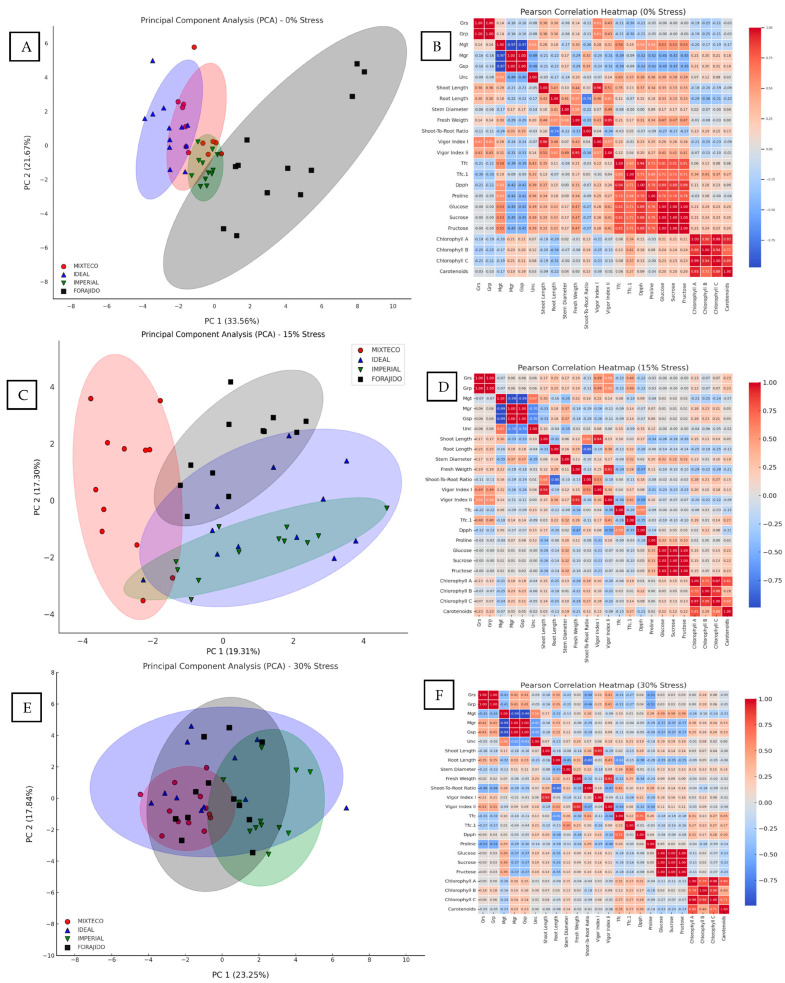
Principal component analysis (PCA) and Pearson correlation heatmaps under different water stress levels. Panels (**A**), (**C**), and (**E**) represent the PCA biplots of morphophysiological and biochemical traits evaluated in four jalapeño pepper varieties (MIXTECO, IDEAL, IMPERIAL, and FORAJIDO) under 0%, 15%, and 30% water stress conditions, respectively. The ellipses represent the distribution and clustering of each variety based on the first two principal components. Panels (**B**), (**D**), and (**F**) show the corresponding Pearson correlation heatmaps for the same variables and stress levels. Positive correlations are shown in red and negative in blue, with the intensity indicating the strength of the correlation. These analyses highlight the multivariate relationships and changes in trait associations as water stress intensifies.

**Figure 6 plants-14-01726-f006:**
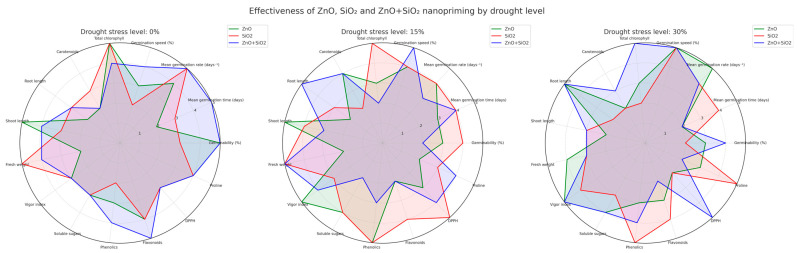
A radar chart comparing ZnO (green), SiO_2_ (red), and ZnO + SiO_2_ (blue) nanopriming in jalapeño under 0%, 15%, and 30% drought stress. Fifteen variables were evaluated. ZnO+SiO_2_ showed the best overall performance under 30% stress, while SiO_2_ stood out under 15%. Combined treatments improved both growth and antioxidant responses under drought.

**Figure 7 plants-14-01726-f007:**
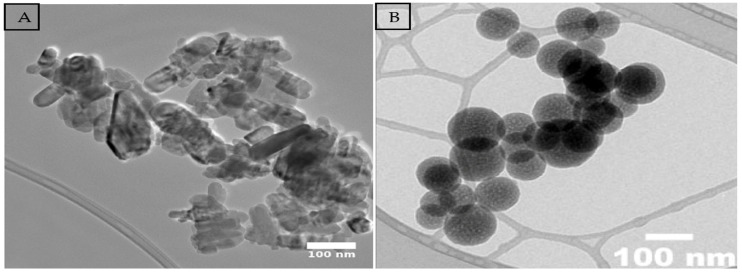
Morphology of samples using transmission electron microscopy (TEM): (**A**) ZnO; (**B**) SiO_2_.

**Figure 8 plants-14-01726-f008:**
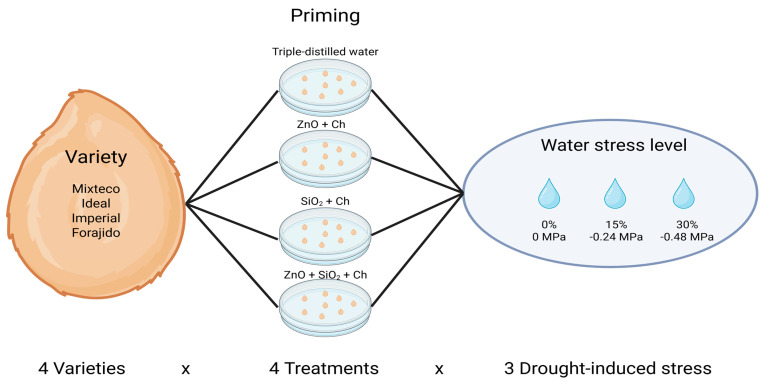
The experimental design with a 4 × 4 × 3 factorial arrangement, evaluating four varieties of jalapeño pepper, four priming treatments, and three levels of induced water stress.

**Figure 9 plants-14-01726-f009:**
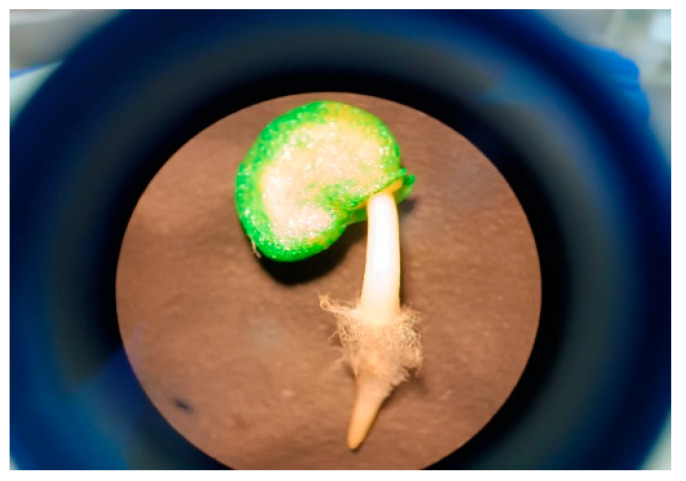
Germination radicle length of 1 to 2 mm (Binocular stereo microscope with VE-S4 zoom system (Velab Co., Pharr, TX, USA)).

**Table 1 plants-14-01726-t001:** Germination parameters of four commercial varieties of jalapeño pepper under different priming treatments.

Drought Stress	Treatment	Variety	Code	Germinated Seeds (n)	Germinability (%)	Mean Germination Time (Days)	Mean Germination Rate (Days^−1^)	Germination Speed (%)	Uncertainty Index (bit)
**0%**	T1 Hydropriming	MT10	7.00 ± 0.00 ^bc^	87.50 ± 0.00 ^bc^	3.334 ± 0.08 ^bcde^	0.300 ± 0.007 ^cde^	30.0 ± 0.7 ^cde^	1.43 ± 0.04 ^a^	MT10
		IT10	6.00 ± 0.00 ^d^	75.00 ± 0.00 ^d^	3.444 ± 0.10 ^bcd^	0.291 ± 0.008 ^de^	29.0 ± 0.8 ^de^	0.97 ± 0.05 ^ab^	IT10
		IMT10	8.00 ± 0.00 ^a^	100.00 ± 0.00 ^a^	3.458 ± 0.07 ^bcd^	0.289 ± 0.006 ^de^	28.9 ± 0.6 ^de^	0.98 ± 0.03 ^ab^	IMT10
		FT10	8.00 ± 0.00 ^a^	100.00 ± 0.00 ^a^	3.458 ± 0.07 ^bcd^	0.289 ± 0.006 ^de^	28.9 ± 0.6 ^de^	0.98 ± 0.03 ^ab^	FT10
	T2 ZnO + Q	MT20	7.00 ± 0.00 ^bc^	87.50 ± 0.00 ^bc^	3.191 ± 0.08 ^cde^	0.313 ± 0.008 ^bcd^	31.4 ± 0.8 ^bc^	1.40 ± 0.04 ^a^	MT20
		IT20	7.00 ± 0.00 ^bc^	87.50 ± 0.00 ^bc^	3.334 ± 0.08 ^bcde^	0.300 ± 0.007 ^cde^	30.0 ± 0.7 ^bde^	0.90 ± 0.07 ^ab^	IT20
		IMT20	8.00 ± 0.00 ^a^	100.00 ± 0.00 ^a^	3.583 ± 0.26 ^bc^	0.280 ± 0.020 ^def^	28.0 ± 2.0 ^def^	1.12 ± 0.25 ^ab^	IMT20
		FT20	8.00 ± 0.00 ^a^	100.00 ± 0.00 ^a^	3.458 ± 0.26 ^bcd^	0.290 ± 0.021 ^de^	29.0 ± 2.1 ^de^	0.86 ± 0.08 ^b^	FT20
	T3 SiO_2_ + Q	MT30	7.00 ± 0.00 ^bc^	87.50 ± 0.00 ^bc^	2.905 ± 0.08 ^e^	0.344 ± 0.010 ^b^	34.4 ± 1.0 ^b^	0.78 ± 0.32 ^b^	MT30
		IT30	7.00 ± 0.00 ^bc^	87.50 ± 0.00 ^bc^	2.334 ± 0.08 ^f^	0.429 ± 0.015 ^a^	42.9 ± 1.5 ^a^	0.90 ± 0.07 ^ab^	IT30
		IMT30	8.00 ± 0.00 ^a^	100.00 ± 0.00 ^a^	3.292 ± 0.07 ^bcde^	0.304 ± 0.007 ^cde^	30.4 ± 0.7 ^cde^	0.86 ± 0.08 ^b^	IMT30
		FT30	7.66 ± 0.57 ^ab^	95.83 ± 7.21 ^ab^	4.137 ± 0.14 ^a^	0.242 ± 0.009 ^g^	24.2 ± 0.8 ^f^	1.42 ± 0.10 ^a^	FT30
	T4 ZnO + SiO_2_ + Q	MT40	7.66 ± 0.57 ^ab^	95.83 ± 7.21 ^ab^	3.000 ± 0.13 ^de^	0.334 ± 0.014 ^bc^	33.4 ± 1.4 ^bc^	1.00 ± 0.40 ^ab^	MT40
		IT40	7.00 ± 0.00 ^bc^	87.50 ± 0.00 ^bc^	3.334 ± 0.08 ^bcde^	0.300 ± 0.007 ^cde^	30.0 ± 0.7 ^cde^	0.90 ± 0.07 ^ab^	IT40
		IMT40	8.00 ± 0.00 ^a^	100.00 ± 0.00 ^a^	3.708 ± 0.26 ^ab^	0.271 ± 0.019 ^efg^	27.1 ± 1.8 ^ef^	1.27 ± 0.23 ^ab^	IMT40
		FT40	6.33 ± 0.57 ^cd^	83.33 ± 7.21 ^cd^	4.095 ± 0.17 ^a^	0.244 ± 0.010 ^fg^	24.4 ± 1.0 ^f^	1.42 ± 0.15 ^a^	FT40
**15%**	T1 Hydropriming	MT115	8.00 ± 0.00 ^a^	100.00 ± 0.00 ^a^	2.917 ± 0.07 ^fg^	0.343 ± 0.009 ^a^	34.3 ± 0.8 ^a^	0.72 ± 0.30 ^ab^	MT115
		IT115	7.33 ± 0.57 ^abc^	91.66 ± 7.21 ^abc^	3.696 ± 0.28 ^abc^	0.271 ± 0.021 ^de^	27.2 ± 2.1 ^de^	1.20 ± 0.16 ^ab^	IT115
		IMT115	7.00 ± 0.00 ^bc^	87.50 ± 0.00 ^bc^	3.952 ± 0.41 ^a^	0.255 ± 0.025 ^e^	25.5 ± 2.5 ^e^	1.25 ± 0.34 ^ab^	IMT115
		FT115	8.00 ± 0.00 ^a^	100.00 ± 0.00 ^a^	3.375 ± 0.13 ^abc^	0.297 ± 0.011 ^bcd^	29.7 ± 1.1 ^bcd^	1.02 ± 0.25 ^ab^	FT115
	T2 ZnO + Q	MT215	7.00 ± 0.00 ^bc^	87.50 ± 0.00 ^bc^	2.905 ± 0.08 ^g^	0.344 ± 0.010 ^a^	34.4 ± 1.0 ^a^	0.78 ± 0.32 ^ab^	MT215
		IT215	7.66 ± 0.57 ^ab^	95.83 ± 7.21 ^ab^	3.476 ± 0.04 ^abcde^	0.288 ± 0.003 ^cde^	28.8 ± 0.3 ^cde^	1.09 ± 0.18 ^ab^	IT215
		IMT215	7.00 ± 0.00 ^bc^	87.50 ± 0.00 ^bc^	3.334 ± 0.08 ^abc^	0.300 ± 0.007 ^bcd^	30.0 ± 0.7 ^bcd^	0.90 ± 0.07 ^ab^	IMT215
		FT215	8.00 ± 0.00 ^a^	100.00 ± 0.00 ^a^	3.375 ± 0.08 ^abc^	0.297 ± 0.007 ^bcd^	29.7 ± 0.7 ^bcd^	0.92 ± 0.07 ^ab^	FT215
	T3 SiO_2_ + Q	MT315	8.00 ± 0.00 ^a^	100.00 ± 0.00 ^a^	3.042 ± 0.07 ^efg^	0.329 ± 0.008 ^ab^	32.9 ± 0.8 ^ab^	0.54 ± 0.53 ^ab^	MT315
		IT315	8.00 ± 0.00 ^a^	100.00 ± 0.00 ^a^	3.208 ± 0.07 ^cdefg^	0.312 ± 0.007 ^abc^	31.2 ± 0.7 ^abc^	0.72 ± 0.15 ^b^	IT315
		IMT315	6.66 ± 0.57 ^cd^	83.33 ± 7.21 ^cd^	3.595 ± 0.11 ^abcd^	0.278 ± 0.009 ^cde^	27.8 ± 0.8 ^cde^	1.36 ± 0.10 ^a^	IMT315
		FT315	7.00 ± 0.00 ^bc^	87.50 ± 0.00 ^bc^	3.429 ± 0.14 ^bcdef^	0.292 ± 0.012 ^bcde^	29.2 ± 1.2 ^bcde^	0.94 ± 0.07 ^ab^	FT315
	T4 ZnO + SiO_2_ + Q	MT415	6.00 ± 0.00 ^d^	75.00 ± 0.00 ^d^	3.167 ± 0.17 ^defg^	0.316 ± 0.017 ^abc^	31.6 ± 1.7 ^abc^	0.52 ± 0.47 ^fb^	MT415
		IT415	8.00 ± 0.00 ^a^	100.00 ± 0.00 ^a^	3.417 ± 0.07 ^bcdef^	0.293 ± 0.006 ^bcde^	29.3 ± 0.6 ^bcde^	0.97 ± 0.03 ^ab^	IT415
		IMT415	8.00 ± 0.00 ^a^	100.00 ± 0.00 ^a^	3.792 ± 0.14 ^ab^	0.264 ± 0.010 ^de^	26.4 ± 1.0 ^de^	1.46 ± 0.09 ^a^	IMT415
		FT415	8.00 ± 0.00 ^a^	100.00 ± 0.00 ^a^	3.458 ± 0.14 ^abcde^	0.289 ± 0.012 ^efg^	28.9 ± 1.2 ^bcde^	1.14 ± 0.24 ^abcde^	FT415
**30%**	T1 Hydropriming	MT130	7.66 ± 0.57 ^ab^	95.83 ± 7.21 ^ab^	3.268 ± 0.15 ^d^	0.307 ± 0.014 ^a^	30.6 ± 1.4 ^a^	1.03 ± 0.25 ^bcd^	MT130
		IT130	8.00 ± 0.00 ^a^	100.00 ± 0.00 ^a^	3.375 ± 0.13 ^cd^	0.297 ± 0.011 ^abc^	29.7 ± 1.1 ^abc^	0.92 ± 0.10 ^d^	IT130
		IMT130	8.00 ± 0.00 ^a^	100.00 ± 0.00 ^a^	4.292 ± 0.19 ^ab^	0.233 ± 0.010 ^e^	23.3 ± 1.0 ^e^	1.82 ± 0.23 ^a^	IMT130
		FT130	7.00 ± 0.00 ^ab^	87.50 ± 0.00 ^ab^	4.095 ± 0.08 ^abc^	0.244 ± 0.005 ^de^	24.4 ± 0.5 ^de^	1.36 ± 0.20 ^abcd^	FT130
	T2 ZnO + Q	MT230	8.00 ± 0.00 ^a^	100.00 ± 0.00 ^a^	3.708 ± 0.07 ^abcd^	0.270 ± 0.005 ^abcde^	27.0 ± 0.5 ^abcde^	1.44 ± 0.05 ^abcd^	MT230
		IT230	6.33 ± 1.52 ^b^	79.16 ± 19.04 ^b^	3.992 ± 0.46 ^abcd^	0.253 ± 0.028 ^cde^	25.3 ± 2.8 ^cde^	1.17 ± 0.18 ^bcd^	IT230
		IMT230	8.00 ± 0.00 ^a^	100.00 ± 0.00 ^a^	4.000 ± 0.33 ^abcd^	0.251 ± 0.020 ^cde^	25.1 ± 2.0 ^cde^	1.45 ± 0.14 ^abc^	IMT230
		FT230	8.00 ± 0.00 ^a^	100.00 ± 0.00 ^a^	3.292 ± 0.07 ^d^	0.304 ± 0.007 ^ab^	30.4 ± 0.7 ^ab^	0.86 ± 0.08 ^d^	FT230
	T3 SiO_2_ + Q	MT330	8.00 ± 0.00 ^a^	100.00 ± 0.00 ^a^	3.625 ± 0.13 ^bcd^	0.276 ± 0.010 ^abcde^	27.6 ± 1.0 ^abcde^	1.12 ± 0.25 ^bcd^	MT330
		IT330	7.33 ± 0.57 ^ab^	91.66 ± 7.21 ^ab^	3.690 ± 0.27 ^abcd^	0.272 ± 0.019 ^abcde^	27.2 ± 1.9 ^abcde^	1.22 ± 0.21 ^bcd^	IT330
		IMT330	7.00 ± 0.00 ^ab^	87.50 ± 0.00 ^ab^	3.857 ± 0.38 ^abcd^	0.261 ± 0.025 ^abcde^	26.1 ± 2.4 ^abcde^	1.56 ± 0.25 ^abc^	IMT330
		FT330	7.33 ± 0.57 ^ab^	91.66 ± 7.21 ^ab^	3.958 ± 0.29 ^abcd^	0.253 ± 0.018 ^bcde^	25.4 ± 1.8 ^bcde^	1.62 ± 0.20 ^ab^	FT330
	T4 ZnO + SiO_2_ + Q	MT430	8.00 ± 0.00 ^a^	100.00 ± 0.00 ^a^	3.417 ± 0.07 ^cd^	0.293 ± 0.006 ^abcd^	29.3 ± 0.6 ^abcd^	0.97 ± 0.03 ^cd^	MT430
		IT430	8.00 ± 0.00 ^a^	100.00 ± 0.00 ^a^	3.500 ± 0.25 ^cd^	0.287 ± 0.021 ^abcd^	28.7 ± 2.1 ^abcd^	1.27 ± 0.23 ^abcd^	IT430
		IMT430	6.33 ± 0.57 ^b^	79.16 ± 7.21 ^b^	4.436 ± 0.36 ^a^	0.226 ± 0.017 ^e^	22.6 ± 1.8 ^e^	1.18 ± 0.24 ^bcd^	IMT430
		FT430	6.33 ± 0.57 ^b^	79.16 ± 7.21 ^b^	3.690 ± 0.13 ^abcd^	0.271 ± 0.010 ^abcde^	27.1 ± 1.0 ^abcde^	1.43 ± 0.05 ^abcd^	FT430

T1: (Hydropriming); T2: (ZnO + Q); T3: (SiO_2_ + Q); T4: (ZnO + SiO_2_ + Q); and water stress levels (0%, 15%, and 30%). Statistical analysis was performed using the Kruskal–Wallis test, followed by Dunn’s multiple comparisons (*p* ≤ 0.05). Superscript letters indicate significant differences.

**Table 2 plants-14-01726-t002:** A description of the distribution of the varieties of jalapeño pepper used, the treatments, the level of induced stress, and the code used in this study.

Jalapeño Pepper Varieties	Treatment NPs + Chitosan	Code		
		Level of Osmotic Stress		
		0%	15%	30%
**Mixteco** **(M)**	(T1) Triple-distilled water Control	MT10	MT115	MT130
	(T2) ZnO 100 mgL^−1^ + Q 100 mgL^−1^	MT20	MT215	MT230
	(T3) SiO_2_ 10 mgL^−1^ + Q 100 mgL^−1^	MT30	MT315	MT330
	(T4) ZnO 100 mgL^−1^ + SiO_2_ 10 mgL^−1^ + Q 100 mgL^−1^	MT40	MT415	MT430
**Ideal** **(I)**	(T1) Triple-distilled water Control	IT10	IT115	IT130
	(T2) ZnO 100 mgL^−1^ + Q 100 mgL^−1^	IT20	IT215	IT230
	(T3) SiO_2_ 10 mgL^−1^ + Q 100 mgL^−1^	IT30	IT315	IT330
	(T4) ZnO 100 mgL^−1^ + SiO_2_ 10 mgL^−1^ + Q 100 mgL^−1^	IT40	IT415	IT430
**Imperial** **(IM)**	(T1) Triple-distilled water Control	IMT10	IMT115	IMT130
	(T2) ZnO 100 mgL^−1^ + Q 100 mgL^−1^	IMT20	IMT215	IMT230
	(T3) SiO_2_ 10 mgL^−1^ + Q 100 mgL^−1^	IMT30	IMT315	IMT330
	(T4) ZnO 100 mgL^−1^ + SiO_2_ 10 mgL^−1^ + Q 100 mgL^−1^	IMT40	IMT415	IMT430
**Forajido** **(F)**	(T1) Triple-distilled water Control	FT10	FT115	FT130
	(T2) ZnO 100 mgL^−1^ + Q 100 mgL^−1^	FT20	FT215	FT230
	(T3) SiO_2_ 10 mgL^−1^ + Q 100 mgL^−1^	FT30	FT315	FT330
	(T4) ZnO 100 mgL^−1^ + SiO_2_ 10 mgL^−1^ + Q 100 mgL^−1^	FT40	FT415	FT430

## Data Availability

The authors declare that all data discussed in this study are available in the manuscript.

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
