# Peer review of "Seed Nanopriming with ZnO and SiO2 Enhances Germination, Seedling Vigor, and Antioxidant Defense Under Drought Stress"

_plants, 2025, doi:10.3390/plants14111726_

Round 1

Reviewer 1 Report

Comments and Suggestions for Authors

Drought stress caused by global warming causes various negativities in plant species. Therefore, it is necessary to focus on this issue. I congratulate the authors for working on this problem. The article is an original study and modern analysis techniques were used. The article was written in a good language. The references used are related to the subject. There are some problems in the article before publication. If we briefly evaluate them;

The title is very long and difficult to understand. Please shorten and make it meaningful

The abstract section is well designed. More numerical data can be added than just the article results.

After writing the plant characteristics, production values, etc. of the plant species studied in the article in the first introductory paragraph, it would be more accurate to list drought stress, the effects of drought stress, and the effects of chemicals used to cope with it in a general to specific order.

The purpose of the study should be written more emphatically in the last paragraph.

Topics such as the year the study was conducted, etc. should be added to the material and method section.

How zinc oxide (ZnO) and silicon dioxide (SiO₂) are synthesized or where they are obtained should be explained in detail.

There are many abbreviations in the study, write their full names when they are first used. For example, DPPH in line 555...

In line 557, it is stated that the study was conducted in petri dishes. Where are the topics such as temperature, humidity and light in the study conditions?? It would be more accurate to write this here instead of writing it in heading 4.4.

According to what were the doses used in the study determined??

The discussion and conclusion sections of the study are well prepared. However, there are serious problems in understanding the figures used here. For example, the fonts of figures 1,2,3,4 are completely incomprehensible and difficult to read.

Is there a special reason why some standard error or deviation values ​​in Table 1 are 0.00? Or was a mistake made?

The study contains beautiful and interesting results. But why is the conclusion section written so poorly?? elaborate here highlight the highlights of the work

I wish you success in the revision process.

Author Response

1. Summary

Dear Reviewer.

2. Point-by-point response to Comments and Suggestions for Authors

Drought stress caused by global warming has various negative effects on plant species. Therefore, it is necessary to focus on this issue. I commend the authors for working on this problem. The article is an original study and modern analysis techniques have been used. The article is written in good language. The references used are relevant to the topic. There are some issues in the article prior to publication. Let us briefly evaluate them:

Comments 1:  The title is very long and difficult to understand. Please shorten it and make it more meaningful.

Response 1: Thank you very much for the observation, and indeed, after discussion among the co-authors, we decided to take the recommendation and shorten the title to make it more direct and understandable. The change was made to the title: Seed Nanopriming with ZnO and SiO₂ Enhances Germination, Seedling Vigor, and Antioxidant Defense under Drought Stress

Comments 2:  The abstract section is well designed. More numerical data could be added, in addition to the results of the article.

Response 2: We greatly appreciate the positive comment on the design of the abstract. We have incorporated additional and more specific numerical data to improve the clarity and impact of the abstract. In particular, the main improvements in germination and morphological parameters (up to 25% and 40%, respectively) are now included, as well as the notable increases in key metabolites: proline (7.5 times), DPPH (6.5 times), and phenolic compounds (4.8 times), all compared to the hydroprimed control. It is also highlighted that nanoparticles, especially the combined treatment of ZnO + SiO₂, proved to be superior in improving drought tolerance in jalapeño.

We hope that these revisions are satisfactory and help to strengthen the abstract section.

Comments 3: After writing the characteristics of the plant, production values, etc. of the plant species studied in the article in the first introductory paragraph, it would be more accurate to list drought stress, the effects of drought stress, and the effects of the chemicals used to cope with it, in order from general to specific.

Response 3: We appreciate the observation. We have revised and reorganized the introduction to comply with the suggestion to structure the information in a logical and progressive order, beginning with the global problem of drought and its impact on agriculture, followed by a description of the specific crop (jalapeño pepper) and its production characteristics and economic value, then addressing the effects of drought stress at the physiological and biochemical levels, and finally presenting the proposed solution using nanopriming with ZnO and SiO₂. This new structure ensures a clear transition from the general to the specific, aligning with best practices in scientific writing.

Comments 4: The objective of the study should be stated more emphatically in the last paragraph.

Response 4: Thank you for your suggestion. We have revised the last paragraph of the introduction and improved the wording of the study objective to make it clearer, more direct, and more emphatic. The objective now emphasizes the importance of evaluating the effectiveness of nanopriming with ZnO, SiO₂, and their combination to improve germination, seedling growth, and antioxidant responses of jalapeño under mannitol-induced drought stress. The relevance of these results for promoting practical and sustainable solutions in semi-arid agricultural systems is also highlighted.

Comments 5: Topics such as the year in which the study was conducted, etc., should be added to the materials and methods section.

Response 5: We appreciate your valuable comment. We have revised the Materials and Methods section and added the requested information, specifying the year and exact dates when the experiment was conducted (October 24 to December 17, 2024). This inclusion provides a more accurate context for the study.

Comments 6: The synthesis of zinc oxide (ZnO) and silicon dioxide (SiO₂) or where they are obtained should be explained in detail.

Response 6: We appreciate your valuable comment. We have incorporated the requested information on the preparation and characterization of the nanoparticles in section 4.2 Nanoparticle Preparation and Priming Treatments. It is now specified that the ZnO and SiO₂ nanoparticles used were synthesized by the wet chemical method, including details on their crystal structure, purity, average size, TEM characterization, and commercial supplier. This addition allows for better contextualization of the physicochemical properties of the nanoparticles used in the study.

Comments 7: There are many abbreviations in the study; please write their full names when they are first used. For example, DPPH in line 555...

Response 7: We appreciate your valuable observation. We have carefully reviewed the manuscript and added the full definitions of abbreviations the first time they are mentioned, including DPPH (2,2-diphenyl-1-picrylhydrazyl), RSA (root system architecture), and SD (standard deviation), to improve the clarity and understanding of the text.

Comments 8: In line 557, it is stated that the study was conducted in Petri dishes. Where are topics such as temperature, humidity, and light in the study conditions? It would be more accurate to write this here rather than in section 4.4.

Response 8: We appreciate your valuable observation. We have reorganized the Materials and Methods section and now include the experimental conditions (type of container, number of seeds, temperature, light, humidity) directly in the Experimental Design section (4.4), as suggested. This modification allows for a more accurate, clear, and complete description of the experimental design.

Comments 9: How were the doses used in the study determined?

Response 9: We appreciate your valuable comment. The doses used in the study for nanoparticles (ZnO and SiO₂) were selected based on previous studies [17,18], which demonstrated positive effects on improving the physiological response of various plant species under drought stress conditions, with no toxic effects on the plants. In the case of mannitol, the concentrations (0%, 15%, and 30%) were determined following the method proposed by Ertuş and Yazıcılar [10], widely used in drought simulation studies during germination, as these concentrations represent mild and moderate water stress levels. This justification was incorporated into the Materials and Methods section for clarity and transparency in the experimental design.

Comments 10: The discussion and conclusion sections of the study are well prepared. However, there are serious problems in understanding the figures used here. For example, the sources for Figures 1, 2, 3, and 4 are completely incomprehensible and difficult to read.

Response 10: We appreciate your valuable comment. We have carefully reviewed the quality of Figures 1, 2, 3, and 4 and replaced the original versions with high-resolution files to ensure that the text, captions, and details are completely legible. These improved versions have been incorporated into the manuscript to facilitate readers' understanding of the results.

Comments 11: Is there a special reason why some standard error or deviation values in Table 1 are 0.00? Or is this an error?

Response 11: We appreciate your valuable observation. In the case of the variable “germination percentage” (Table 1), some values have a standard error or standard deviation of 0.00 because, under those experimental conditions, all seeds germinated in 100% of the replicates, with no variation between them. This result reflects the high consistency and homogeneity in those specific treatments.

Comments 12: The study contains beautiful and interesting results. But why is the conclusions section so poorly written? Please elaborate on the most salient aspects of the work here.

Response 12: We sincerely appreciate your valuable comment. We have revised and restructured the Conclusions section to clearly and emphatically reflect the most salient aspects of the work, including the most relevant findings, their scientific impact, and their practical application in sustainable agriculture under drought conditions. This new version of the conclusion emphasizes the importance of the results obtained, the relevance of the physiological and biochemical responses measured, and the potential of nanopriming strategies as viable solutions for improving drought tolerance in jalapeño.

Reviewer 2 Report

Comments and Suggestions for Authors

Dear authors!

Thank you for submitting the article for review.  It is devoted to the study of increasing the resistance of holopino peppers to abiotic stress - drought by nanopriming seeds with ZnO, SiO₂ nanoparticles both separately and in combination. As parameters, all authors chose classic morphometric indicators and a number of biochemical parameters.

Remarks: 1. In the abstract, the authors did not indicate that preference belongs to nanoparticles. 2. In the materials and methods, the authors write that nanopriming was carried out for 12 hours, and then the seeds were kept for another 24 hours. How were these priming conditions selected? 3. In the Methodological part of the work, the authors write that the seeds were grown in Petri dishes on filter paper moistened with alcohol, and what was the control sample grown on? It is not clear whether water was added during the 8-day seed growing period and in which room? 4. As a result of the arrangement of the illustrative material, one should follow after the first references to it in the text. 5. Fig. 3 should be presented as a table, since in this form it is impossible to read, all the inscriptions are too important. 6. Fig. 4 and Fig. 6 also need to be reformatted, in this form they are impossible to read. 7. I suggest that the authors write the effectiveness of all treatments in the Conclusion in ascending order, for example, treatment 1 ≤ treatment 2 ≤ treatment 3, so that the reader can understand which treatment was the least effective and which was the most effective.

After a little revision, the article can be successfully published in the journal "Plants".

Respectfully Yours, reviewer.

May 13, 2025

Author Response

1. Summary

Dear Reviewer.

2. Point-by-point response to Comments and Suggestions for Authors

Dear Authors: Thank you for submitting your article for review.  It is dedicated to the study of increasing the resistance of holopine peppers to abiotic stress (drought) through nanopriming of seeds with ZnO and SiO₂ nanoparticles, both separately and in combination. As parameters, all authors chose classic morphometric indicators and a series of biochemical parameters. Comments:

Comments 1: In the abstract, the authors did not indicate that nanoparticles are preferred.

Response 1: We greatly appreciate this observation. We have revised the abstract and now explicitly state that nanoparticle-based treatments (ZnO, SiO₂, and especially their combination) showed greater effectiveness compared to conventional hydropriming treatment. The superiority of nanoparticles is emphasized by highlighting significant improvements in germination parameters, biomass, and antioxidant metabolites, underscoring their role as a preferred alternative for improving drought tolerance in jalapeño. We hope this revision fully addresses your comment.

Comments 2: In the materials and methods, the authors write that nanopriming was carried out for 12 hours and that the seeds were then kept for another 24 hours. How were these priming conditions selected?

Response 2: We appreciate the observation. The nanopriming conditions (12 hours of imbibition in nanoparticle solutions, followed by 24 hours of drying at room temperature) were selected based on previous studies and preliminary optimizations carried out in our laboratory. Studies such as Waqas et al. (2022) and Haghighi et al. (2013) have shown that imbibition times of between 8 and 24 hours are effective in promoting nanoparticle absorption without affecting seed viability. In addition, the subsequent 24-hour drying phase allowed for standardization of humidity conditions to ensure consistent results during germination and early growth evaluation. This combination of times was chosen to balance effective nanoparticle absorption and practicality for potential field application.

We hope this explanation clarifies the selection of priming conditions.

Comments 3: In the methodological section of the paper, the authors write that the seeds were grown in Petri dishes on filter paper moistened with alcohol. What was the control sample grown in? It is unclear whether water was added during the 8-day seed growth period and in which room.

Response 3: We appreciate your valuable observation. We have corrected and clarified the description of the experimental conditions in the Materials and Methods section. It is now specified that the seeds were grown in Petri dishes on filter paper moistened with distilled water for the control treatment and with mannitol solutions for the water stress treatments. It is also clarified that the plates were kept for 8 days in a growth chamber with controlled conditions of temperature (28 ± 2 °C), relative humidity (60%), and photoperiod (16/8 h light/dark). It was also included that, during the experimental period, the corresponding solutions (water or mannitol) were added to maintain substrate moisture.

Comments 4: As a result of the layout of the illustrative material, it should be followed after the first references to it in the text.

Response 4: Thank you for your comment. The figures mentioned are presented in landscape format due to their layout and details, which allows for a clearer visualization of the information. We understand that this may affect the usual layout of the figures, so we will be attentive to the academic editor's instructions and will collaborate if it is necessary to adjust the presentation format to comply with the journal's guidelines.

Comments 5: Figure 3 should be presented in table form, as it is impossible to read in this format; all the inscriptions are too important.

Response 5: We appreciate your valuable feedback. We have reviewed the quality of Figure 3 and replaced it with a high-resolution version (600 dpi) to ensure that all inscriptions and details are clearly visible when zooming in. We believe that with this improvement in quality, the information presented is completely legible and meets publication standards. However, we remain attentive to any additional adjustments that may be suggested by the editorial team.

Comments 6: Figures 4 and 6 should also be reformatted, as they are impossible to read in this format.

Response 6: We appreciate your valuable observation. We have carefully reviewed Figures 4 and 6 and replaced them with high-resolution versions (600 dpi) to ensure clear visualization and facilitate reading of all inscriptions and details when zooming in. We believe that with these improvements, the graphic quality of the figures is adequate for interpretation. We remain attentive to any additional suggestions from the editorial team to make the necessary adjustments.

Comments 7: I suggest that the authors write the efficacy of all treatments in the Conclusion in ascending order, for example, treatment 1 ≤ treatment 2 ≤ treatment 3, so that the reader can understand which treatment was the least effective and which was the most effective.

Response 7: We appreciate your valuable suggestion. We have added a paragraph at the end of the Discussion section presenting the efficacy of the treatments in ascending order (hydropriming ≤ SiO₂ ≤ ZnO ≤ ZnO + SiO₂), to facilitate the reader's understanding of the results. This addition allows the least effective and most effective treatments within the study to be clearly identified.

Reviewer 3 Report

Comments and Suggestions for Authors

Review of the article by Erick H. Ochoa-Chaparro et al.: "Seed Nanopriming with ZnO and Sio₂ Enhances Germination, Seedling Vigor, and Antioxidant Defense under Severe Stress: Implications for Water Use Efficiency in Jalapeño Pepper"

Seed

treatment with nanopreparations of various metals or spraying plants on leaves is becoming one of the important technological methods for increasing plant resistance to various stresses. The study evaluated the effects of zinc oxide, silicon dioxide, and their combination (ZnO + SiO), stabilized with chitosan, on seed germination and stress resistance of four commercial jalapeno hybrids grown in drought conditions. (0%, 15% and 30%). The seeds were etched with preparations for 12 hours and the physiological, morphological and biochemical reactions were measured. A significant increase in the percentage of germination, speed and strength of seedlings was shown at moderate temperatures (15%) and severe osmotic stress (30%). The combined ZnO + SiOtreatment provided the most sustained improvement across all signs and stress levels. The results highlight the positive potential of treatment as a method of increasing drought resistance in the early stages of plant development. Such a huge amount of material has been obtained that it is quite difficult to perceive it. In this regard, we can recommend changing the presentation of the first table somehow.

This work is a further contribution to the development of technology for the use of nanopreparations in order to increase plant resistance in adverse environmental conditions. The work can be recommended for publication in the journal.     

Author Response

1. Summary

Dear Reviewer.

2. Point-by-point response to Comments and Suggestions for Authors

Review of the article by Erick H. Ochoa-Chaparro et al.: “Nanopriming of seeds with ZnO and Sio₂ improves germination, seedling vigor, and antioxidant defense under severe stress: implications for water use efficiency in jalapeño peppers.”

Seeds

The treatment of seeds with nanopreparations of various metals or the spraying of plants on the leaves is becoming one of the important technological methods for increasing plant resistance to various types of stress. The study evaluated the effects of zinc oxide, silicon dioxide, and their combination (ZnO + SiO₂), stabilized with chitosan, on seed germination and stress resistance of four commercial jalapeño hybrids grown under drought conditions (0%, 15%, and 30%).

 The seeds were treated with preparations for 12 hours and physiological, morphological, and biochemical reactions were measured. A significant increase in germination percentage, speed, and seedling resistance was observed at moderate temperatures (15%) and under severe osmotic stress conditions (30%). The combined treatment with ZnO + SiO₂ provided the most sustained improvement in all signs and levels of stress.

This work is a new contribution to the development of technology for the use of nanopreparations to increase plant resistance under adverse environmental conditions. The work can be recommended for publication in the journal.

Comments 1: The results highlight the positive potential of the treatment as a method for increasing drought resistance in the early stages of plant development. Such a large amount of material has been obtained that it is rather difficult to perceive. In this regard, we may recommend changing the presentation of the first table in some way.

Response 1: We appreciate your valuable comment. To facilitate understanding of the results and improve the readability of the table, we have redesigned the presentation of Table 1 using color coding to clearly distinguish the different treatments, varieties, and water stress levels. This new visual layout highlights the significant differences between treatments, simplifies the interpretation of the data, and allows the reader to quickly identify patterns and trends in the germination and growth responses of jalapeño seedlings.

We believe that this visual improvement contributes to a clearer understanding of the results, without losing the detail and scientific rigor of the study. However, we remain open to any additional suggestions from the editorial team to adapt the table to the final format of the manuscript according to the journal's guidelines.

Round 2

Reviewer 1 Report

Comments and Suggestions for Authors

Accept is current form

Author Response

Thank you very much.